# Psychological Distress and Physical Adverse Events of COVID-19 Vaccination among Healthcare Workers in Taiwan

**DOI:** 10.3390/vaccines11010129

**Published:** 2023-01-05

**Authors:** Ahmad Rifai, Wan-Ching Wu, Yu-Wen Tang, Mei-Yun Lu, Pei-Jen Chiu, Carol Strong, Chung-Ying Lin, Po-Lin Chen, Wen-Chien Ko, Nai-Ying Ko

**Affiliations:** 1Department of Nursing, College of Medicine, National Cheng Kung University, Tainan 701, Taiwan; 2International Doctoral Program in Nursing, Department of Nursing, College of Medicine, National Cheng Kung University, Tainan 701, Taiwan; 3Faculty of Nursing, University of Jember, Jember 68121, Indonesia; 4Center for Infection Control, National Cheng Kung University Hospital, Tainan 701, Taiwan; 5Department of Public Health, College of Medicine, National Cheng Kung University, Tainan 701, Taiwan; 6Institute of Allied Health Sciences, College of Medicine, National Cheng Kung University, Tainan 701, Taiwan; 7Department of Occupational Therapy, College of Medicine, National Cheng Kung University, Tainan 701, Taiwan; 8Biostatistics Consulting Center, National Cheng Kung University Hospital, College of Medicine, National Cheng Kung University, Tainan 701, Taiwan; 9Department of Internal Medicine, National Cheng Kung University Hospital, College of Medicine, National Cheng Kung University, Tainan 701, Taiwan

**Keywords:** COVID-19, vaccination, adverse events, psychological distress, preventive behaviors, healthcare workers

## Abstract

The COVID-19 pandemic places high pressure on everyone, including healthcare workers (HCWs), thus causing them to experience psychological distress. HCWs have priority in receiving the COVID-19 vaccine. However, few studies have identified adverse events (AEs) and psychological distress in the HCWs group. Therefore, we investigated the association between psychological distress and AEs and the determinants of protective behaviors in Taiwanese HCWs. A longitudinal measurement was conducted among HCWs at National Cheng Kung University Hospital (NCKUH), Tainan, Taiwan (n = 483, mean age = 37.55 years). All HCWs completed an online questionnaire on psychological distress, COVID-19 vaccination AEs, and protective behaviors. We used generalized estimating equations (GEE) to analyze the correlation between psychological distress and AEs, and used multivariable logistic regressions to explore the predictors of protective behaviors. Depression and distress and anger were significantly associated with various physical AEs (*p* = 0.045 to *p* < 0.001). Suicidal thoughts became a significant independent variable of systemic AEs after COVID-19 vaccination (*p* = 0.014 to *p* < 0.001). People of older ages or females engaged more in washing their hands, wearing masks, and reducing their presence in crowded places. Suicidal thoughts were related to the occurrence of systemic AEs among HCWs. Doctors performed better at preventive behaviors compared to nurses and other HCWs. HCWs who experienced anxiety and nervousness tended to avoid crowds.

## 1. Introduction

The prevalence of COVID-19 among frontline healthcare workers (HCWs) was 2747 cases per 100,000 compared with 242 cases per 100,000 people in the general community during the first month of the COVID-19 pandemic in the UK and the US [1]. This figure indicates that HCWs have a 10-fold risk of COVID-19 infection because of direct patient care. Moreover, direct patient care during the COVID-19 pandemic, regardless of whether the patient was infected with COVID-19, results in HCWs experiencing both physical and psychosocial symptoms [2]. HCWs have been found to have poor general health, reduced sleep quality, increased psychological distress (e.g., stress, anxiety, and depression), reduced financial income, and decreased family relationships [3,4,5,6,7].

HCWs are the top priority for vaccination to protect them from COVID-19 infection [8]. They have a significant responsibility to promote vaccination and guide patients. On the other hand, they are at high risk of being infected and are potential candidates for spreading the disease [9]. COVID-19 infections among HCWs directly affects their work-related environment and the whole healthcare system. Secondary exposures, isolation, and staff infections can significantly harm a single ward’s capacity to care for patients, creating a multiplier effect on the facility’s functional resilience and staff confidence [10]. Comprehensive protection for HCWs from COVID-19 infection will guarantee the sustainability and protection of the healthcare system [11].

Unfortunately, the prevalence of COVID-19 vaccination hesitancy among HCWs worldwide ranges from 4.3 to 72%. Most studies have found concerns about vaccine safety, efficacy, and potential side effects as the leading reasons for COVID-19 vaccination hesitancy in HCWs [12,13]. Some HCWs reported that they wanted to wait until the vaccine experiences of others were known, and some others stated that they did not trust the rushed Food and Drug Administration (FDA) process [14]. The hesitancy of vaccination uptake among HCWs led to a low rate of COVID-19 vaccine uptake. The Kaiser Family Foundation and the Washington Post (2021) reported that only 52% of US frontline HCWs have received the COVID-19 vaccination. The 48% of HCWs who remained unvaccinated expressed that they had not yet decided if they will accept a vaccine dose. In Taiwan, the HCWs’ willingness to receive the COVID-19 vaccination is lower (23.4%) compared to hospital outpatient visitors (30.7%). The low willingness for COVID-19 vaccine uptake in Taiwan may be due to Taiwan’s status, as it was safer than other countries during the survey period [13].

One of the main reasons for vaccination hesitancy is the adverse events (AEs) after vaccine uptake. The literature has revealed some AEs following COVID-19 vaccination, which differs among various vaccine types. One of the most common vaccines approved by countries is the AstraZeneca vaccine (ChAdOx1 nCoV-19). The frequent AEs incurred by this vaccine are injection site pain (77.7–88.0%), fatigue (50.7–92.9%), myalgia (60.5–80.8%), malaise (83.8%), headaches (47.4–72.0%), and fever (≥38.0 °C, 36.1–38.7%). The severity of most AEs were mild-to-moderate, and occurred less in the older age groups [15,16,17]. Another common vaccine (Pfizer–BioNTech COVID-19) triggered common AEs including injection site pain (89.8%), fatigue (62.2%), headaches (45.6%), muscle pain (37.1%), and chills (33.9%) [18].

Apart from AEs, mental health problems (e.g., acute stress disorder, anxiety, and depression) from COVID-19 among HCWs are important concerns as HCWs are key individuals in the fight against the COVID-19 pandemic. Several systematic reviews have reported that HCWs experienced psychological disorders during the COVID-19 pandemic, including anxiety, depression, stress, sleep disorders, mental health-related factors, and decreased mental well-being [19,20,21,22]. However, to the best of our knowledge, there is a dearth of evidence regarding the AEs and mental health problems after vaccination in Taiwan. This study aimed to investigate the correlation of psychological distress during the pandemic and physical AEs among HCWs after receiving a COVID-19 vaccination. A secondary aim was to describe protective behavior and its association with psychological distress among HCWs.

## 2. Materials and Methods

### 2.1. Study Design and Sampling

This study adopted a longitudinal design and recruited participants from the National Cheng Kung University Hospital (NCKUH), Tainan, Taiwan. NCKUH played a crucial role in combating the spread of COVID-19 by establishing safe and high-quality outdoor coronavirus testing stations during the outbreak, both for patients and HCWs. In the data collection process, researcher 2 (W.-C.W.), researcher 3 (Y.-W.T.), researcher 4 (M.-Y.L.), and researcher 5 (P.-J.C.) approached each HCW who received the COVID-19 vaccination in the NCKUH. The researcher explained the research objectives and procedures, provided a QR code via cell phone, and printed a paper to be scanned by the HCWs. Then, the HCWs scanned a QR code to complete an online survey using Google documents. The first page of the online survey clearly indicated the study’s purpose and the participant’s rights: when an HCW agreed to participate after reading all the information, he or she must hit the “*agree* button” on this page to continue the survey. Those who did not hit the “*agree* button” had their online survey terminated immediately. The data collection was performed from March 24 to April 24, 2021. The inclusion criteria for the eligible participants were: (1) an HCW in the NCKUH; (2) aged over 20 years; and (3) willing to participate in the study.

### 2.2. Measures

#### 2.2.1. Participants’ Characteristics

The online survey collected the following information from the participants: age (fill in a number with the unit of the year); gender (male or female); and type of occupation (physician, nurse, or others).

#### 2.2.2. Physical Adverse Events

The physical AEs were collected by an online survey which included 20 items, including local pain, fatigue, weakness, muscle pain, local swelling, headache, dizziness, chills, local mass, joint pain, fever, local bruise, nausea, vomiting, swollen lymph nodes, poor appetite, abdominal pain, sweating, itching, and skin rash. One additional question (other symptoms) was added to the online survey to cover the other effects experienced by the HCWs. The participants reported their physical AEs five times after they were vaccinated: on day 1, day 2, day 3, day 7, and day 14 after vaccination uptake.

#### 2.2.3. Psychological Distress

We used the Brief Symptom Rating Scale (BSRS-5) [23] to identify the psychological status of HCWs on the day of vaccination. It consisted of five questions identifying the following psychological distress: (i) trouble falling asleep; (ii) feeling tense or high strung; (iii) feeling irritable or angry; (iv) feeling down or depressed; and (v) feeling inferior to others. Each item was measured using a five-point Likert scale (0 = not at all, 1 = a little bit, 2 = moderately, 3 = quite a bit, and 4 = extremely). The assessment of this questionnaire used the following conditions: a score of 0–5 is good emotionally, 6–9 has mild emotional distress, 10–14 is moderate emotional distress, and a score of 15–20 is severe emotional distress. Regarding an additional question about suicidal thoughts, if the score was >2, the person requires expert consultation and psychiatric therapy. The Cronbach’s alpha coefficient for this questionnaire was 0.84 [24].

#### 2.2.4. Protective Behaviors

The Preventive COVID-19 Infection Behaviors Scale (PCIBS) [25] consists of five items as follows: (i) avoiding crowds as much as you can; (ii) keeping your house ventilated; (iii) sanitizing and cleaning your house; (iv) washing your hands as much as you can; and (v) wearing a face mask as much as you can and was used to measure the HCWs behavior to prevent being infected by COVID-19. There were three options in the PCIBS for this study: (i) No, (ii); Yes, because of COVID-19; and (iii) Yes, but not because of COVID-19.

### 2.3. Statistical Analysis

Descriptive statistics were used to analyze the participants’ characteristics including frequency (percentage) for categorical data (e.g., gender) and means ± SD for continuous data (e.g., age). We used generalized estimating equations (GEE) with a linear scale function and an exchangeable structure correlation matrix to analyze the relationship between psychological disorders and physical AEs after receiving the COVID-19 vaccine. The exchangeable structure correlation matrix assessed the correlations between the repeated measures and a small number of missing data could be accommodated for in this model. In the GEE, the repeated measured AEs were treated as dependent variables and the psychological disorders were treated as independent variables. This seemed to be the best option given the fact that there were different trends in the different times of measurement. The relationship between variables with a *p*-value score of less than 0.05 indicated a statistically significant relationship. Then, multivariate logistic regression models were used to examine the adjusted odds ratios (AORs) regarding how different variables such as age, gender, occupation, and psychological distress, explained the participants’ protective behaviors to prevent the COVID-19 infection. All statistical analyses were performed using IBM SPSS 20 (IBM Corp., Armonk, NY, USA).

## 3. Results

There were 5948 HCWs registered with the NCKUH when the study was conducted. During the study period (24 March to 24 April 2021, and the survey continued after this survey), 10.9% of HCWs received the ChAdOx1 nCoV-19 vaccine, and 75.6% of HCWs filled out the online questionnaire. After excluding those with missing information in the measurements of AEs, data from 483 HCWs were used for data analysis. Female HCWs slightly dominated in this study (53.8%); compared to the population, the proportion of female HCWs was 82%. Based on age, most health workers were 30–39 years old (35.4%) with a mean (SD) age of 37.55 (10.73) years, compared to a mean age of the general HCW population of 34.8 years. Doctors were the dominant health profession in this study (38.7%) (Table 1).

There was a significant difference in the physical AEs’ response to each measurement by HCWs. On the first measurement day, 89.86% of respondents experienced AEs. The incidence of AEs continued to decrease significantly on the second, third, seventh, and fourteenth days of measurement after being given the AZ vaccination, at 83.02%, 74.53%, 72.67%, and 63.77%, respectively. HCWs reported that the most common physical AEs after COVID-19 vaccination were local pain, fatigue, weakness, muscle aches, headaches, chills, dizziness, swelling at the injection site, joint pain, poor appetite, and fever (Figure 1). The incidence of physical AEs felt by HCWs existed until the second and third days after vaccination and, on average, experienced a significant decrease on the seventh and fourteenth days of measurement. For example, 89.63%, 89.28%, and 80.00% of respondents reported experiencing local pain on the first, second, and third days of measurements, respectively. On the seventh and fourteenth days of measurement, there was a significant decrease to 26.78% and 9.74% of respondents who reported the incidence of local pain. Fever was felt by 35.25% of respondents on the first day after vaccination and still felt by 20.95% of respondents on the second day of measurement. On the third day, there was a decrease in the incidence of fever to 2.50% of HCWs, and it continued to decrease on the seventh and fourteenth days of measurement.

There was a significant difference between male and female HCWs who experienced AEs (*p* = 0.038 to *p* = 0.001). Similarly, in the age comparison group, there was a significant difference in the incidence of AEs (*p* = 0.002 to *p* ≤ 0.001), except for the incidence of local swelling (*p* = 0.094). There was no significant difference between the type of health professionals and the incidence of AEs in terms of muscle pain, fatigue, weakness, and dizziness (*p* = 0.058 to *p* = 0.103) after receiving the COVID-19 vaccine.

The results of the GEE analysis showed that depression and upset was significantly associated with various physical AEs in the HCWs group (*p* = 0.045 to *p* < 0.001). The type of psychological distress in the form of sleeping disorder only significantly correlated with three AEs, namely fatigue (*p* = 0.001), muscle pain (*p* = 0.049), and headache (*p* = 0.036). Suicidal thoughts became a significant independent variable on the incidence of AEs after COVID-19 vaccination (*p* = 0.014 to *p* < 0.001). All type of psychological distress were correlated with fatigue and muscle pain (*p* = 0.049 to *p* < 0.001) (Table 2) (Appendix A
Table A1).

Of the five mental health domains from the BSRS-5 questionnaire that HCWs had filled out, most experienced mild anxiety disorders (31.7%), mild sleep disorders (30.2%), irritability (24.2%), mild depression (18.8%), and low self-esteem (15.5%). Regarding COVID-19 prevention behaviors among HCWs, the results showed that numerous HCWs did not adhere to health protocols properly: 36.9% of HCWs did not avoid crowds; 55.3% of HCWs did not keep the house clean; 40.4% of HCWs washed hands regularly but not because of COVID-19; and 55.9% of HCWs maintained indoor air ventilation but not because of COVID-19.

The results of multiple logistic regression models related to preventive behaviors showed that female HCWs tended to be more obedient in washing their hands regularly (AOR = 1.08; 95% CI = 0.65–1.77) and wearing masks (AOR = 1.01; 95% CI = 0.57–1.77) when compared to male HCWs. In addition, nurses performed fewer preventive behaviors than doctors and other HCWs (AOR = 0.54–0.99) in almost all aspects except for avoiding crowds (AOR = 1.15; 95% CI = 0.72–1.84). HCWs who experienced nervousness and anxiety tended to avoid crowds 1.83 times more than HCWs who did not feel nervousness and anxiety. Older HCWs were inclined to wash their hands and wear masks (0.97 and 0.96 times more often, respectively, for one year older) (Table 3) (Appendix A
Table A2).

## 4. Discussion

The current study showed that the AEs experienced by HCWs after receiving the first dose of COVID-19 vaccine were local AEs (i.e., pain at the injection site and local swelling) and systemic AEs (i.e., fatigue, weakness, muscle pain, headache, chills, dizziness, joint pain, poor appetite, and fever). However, none of the HCWs reported life-threatening AEs after vaccination. All components of the BSRS-5: sleep disorder, nervousness and anxiety, distress and anger, depression and upset, inferiority to others, and suicidal thoughts, were significantly associated with the incidence of fatigue and muscle pain after COVID-19 vaccination in HCWs. Although depression was significantly associated with all types of AEs, sleep disorders in HCWs were only significantly associated with fatigue, muscle pain, and headaches. Suicidal thoughts were not significantly related to the occurrence of local AEs, namely local pain and swelling. Fever was the only occurrence of AEs associated with one component of psychological distress, namely depression. Being female and of older age were predictors of preventive behaviors, especially washing hands and wearing masks regularly in the HCWs group. An interesting finding was that the nursing profession did not perform preventive behaviors well compared to doctors and other HCWs. Nervousness and anxiety were significant predictors for HCWs in avoiding crowds.

The younger and female participants had a greater chance of experiencing AEs. This study’s results align with results reported by studies across San Francisco, Korea, and Japan [10,15,26]. One of the potential reasons is that the COVID-19 vaccine had a stronger immune response in women and younger people [27]. Another potential explanation is that the antibody response to the COVID-19 vaccine was related to sex hormones for both women and men, but had a greater impact on women [28].

This study additionally identified the occurrence of psychological distress experienced by HCWs during the COVID-19 pandemic. About one-fifth to one-third of the HCWs in the present study experienced different types of psychological distress such as sleep disorders, anxiety, and depression. A few HCWs reported suicidal thoughts (2.9%). The results of this study are consistent with findings from studies on Taiwan frontline HCWs [29], Singaporean HCWs (nurses, doctors, allied health professionals, and administrative and operations staff) [30], African HCWs including physicians, nurses, radiographers, midwives, psychosocial counselors, pharmacists, and medical laboratory technicians [19], and Canadian physicians [31]. Although our paper shared the same studied population of the paper [29], there are major differences between the current study and the published paper by Lu et al. Specifically, the purposes of the two studies are clearly different: the present paper aimed to investigate adverse effects of the COVID-19 vaccine, while Lu et al. aimed to examine psychological distress among healthcare workers who did not receive a vaccination against COVID-19.

Similar findings are supported and have been explained by several systematic reviews. A systematic review reported that HCWs who experience depression, anxiety, stress, sleep problems, mental health-related factors, and decreased mental well-being might be triggered by fear of COVID-19 [20]. Another explanation is that sleep problems among HCWs can potentially increase psychological distress [21]. During the pandemic, HCWs had to keep working and present physically to workplaces such as hospitals and clinics, workplaces which have a high risk of being infected with the novel coronavirus (nCoV). Furthermore, HCWs have a very high workload and face stigmatization from family members and the surrounding environment [20,32]. Therefore, it is essential to take preventive actions as early as possible, so that HCWs will not experience psychological distress. The long-term existence of psychological distress will reduce job performance, job satisfaction and lead to an increase in absenteeism and job turnover among HCWs [33].

The GEE analysis results showed that the presence of fatigue and muscle pain after the first vaccination shot was related to the occurrence of psychological distress reported by HCWs before the vaccination day. A study by Rakel stated that the main complaint of people who were depressed was of fatigue, as well as other symptoms such as headaches, dizziness, palpitations, abdominal cramping, loss of appetite, and pain [34]. HCWs who experienced sleep disturbances during the pandemic had a higher chance of experiencing fatigue, muscle pain, and headaches. An interesting finding from this study is that HCWs who experienced suicidal thoughts had a greater chance of experiencing post-vaccination systemic AEs, compared to HCWs who felt other psychological disorders. The results imply that suicidal thoughts have become a serious psychological disorder because of their high risk of generating individuals who may attempt to take their own life [35]. Although controlling and reducing the problem of suicidal thoughts is difficult, several alternative actions can be taken to prevent HCWs from committing suicide. One is to prepare a safety plan through warning signs, internal coping mechanisms, social contacts, relatives’ help, and professional therapies [35]. Another action that can be engaged to prevent suicidal attempts is to overcome depression experienced by HCWs, as depression is significantly associated with the presence of suicidal thoughts in HCWs [36].

Regarding the implementation of protective behaviors, our results indicated that HCWs who were older or female engaged more in the following behaviors: washing their hands, wearing masks, and avoiding crowded places. The results of this study are consistent with studies conducted in the US which indicated that older adults practice social distancing, mask-wearing, and handwashing at a high level (over 95%) [37]. Similarly, studies in Turkey and Bangkok showed that 70.8% of older adults strictly adhered to preventive behaviors [38,39]. The explanation for the higher levels of preventive behaviors among older HCWs can be due to vulnerability among older people; older people who are infected by the SARS-CoV-2 virus could have rapid disease progression with severe manifestations because of their decreased immunity, leading to a critical condition [40,41].

In addition, nurses in this study did not show the expected preventive behaviors, whereas doctors and other health workers were better at implementing preventive behaviors. A study in India reported the same findings. All doctors involved in the study carried out the actions of wearing masks, maintaining social distancing, and hand washing in good numbers [42]. The number of nurses in the hospital is the highest compared to other health workers. Nurses have a crucial role in health services and can be said to be in direct contact with patients for 24 h. Consequently, nurses sometimes experience fatigue and ignore standard healthcare procedures.

Depressive conditions, low self-esteem, and suicidal thoughts were not shown to be significantly related to preventive behaviors in this study. However, this does not mean that this situation can be ignored, considering that there are still many HCWs who report experiencing these conditions. Sleep disorders and nervousness and anxiety were significantly related to HCWs’ crowd-avoiding behaviors. However, they were not significantly related to measures such as maintaining room ventilation, keeping the house sanitized, washing hands, and wearing masks. Distress was significantly related to the behavior of washing hands regularly. Limkunakul et al. reported that the knowledge and perception of the personal preventability of HCWs in Thailand were related to preventive behaviors [43]. The additional information that components from the health behavioral model also contributes to explaining the factors that encourage HCWs to perform preventive behaviors.

Some limitations of the study are presented as follows. First, some HCWs did not report AEs experienced within the specified time, resulting in missing data. However, data analysis using GEE can handle missing data so that it does not affect the results of the data analysis. Second, the possibility of the occurrence of ascertainment bias may still exist due to the likelihood of some HCWs who agreed to get vaccination voluntarily, compared to those who may reject immunization. Third, this study involved HCWs from one medical center in southern Taiwan, so it cannot be generalized to all areas of Taiwan. There was no significant difference between HCWs in terms of characteristics from the northern and southern parts of Formosa Island. However, future research could involve more HCWs from all regions in Taiwan. The fourth limitation is that during the data collection period, only one type of vaccine was administered to HCWs, the Oxford–AstraZeneca vaccine, so we could not identify a difference in the occurrence of AEs from other types of COVID-19 vaccines. Fifth, the implementation of preventive behaviors was identified only on the first day, when HCWs received their vaccine; therefore, there was not enough information on the consistency of the HCWs’ implementation of preventive behaviors post-vaccination.

## 5. Conclusions

The results of the current study indicate that HCWs experienced various AEs, both local and systemic. Sleep disorders, nervousness and anxiety, distress and anger, depression and upset, inferiority to others, and suicidal thoughts were significantly associated with fatigue and muscle pain after the first dose of COVID-19 vaccination. We also found that suicidal thoughts were related to the occurrence of systemic AEs among HCWs. Doctors performed better at preventive behaviors compared to nurses and other HCWs, while older and female HCWs showed greater frequency in practicing hand washing and wearing masks. HCWs experiencing anxiety and nervousness about COVID-19 were motivated to avoid crowds.

## Figures and Tables

**Figure 1 vaccines-11-00129-f001:**
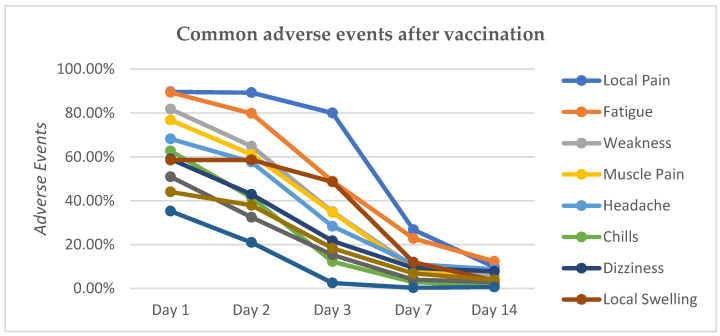
Trend of physical adverse events among HCWs after COVID-19 vaccination on day 1, day 2, day 3, day 7, and day 14 of measurement.

**Table 1 vaccines-11-00129-t001:** Participant characteristics of COVID-19 vaccine recipients.

Variables	ChAdOx1 nCoV-19 Vaccine (n = 483)
Sex Male Female	223 (46.17) 260 (53.83)
Age (mean ± SD)	37.55 ± 10.73
20–29 years old	137 (28.4)
30–39 years old	171 (35.4)
40–49 years old	99 (20.5)
50–59 years old	56 (11.6)
60–69 years old	18 (3.7)
70–79 years old	2 (0.4)
Type of occupation	
Physician	187 (38.7)
Nurse	147 (30.4)
Others	149 (30.8)

Data represent the number (%) of vaccine recipients. SD = standard deviation.

**Table 2 vaccines-11-00129-t002:** Generalized estimating equations for psychological disorders (PDs) and adverse events (AEs).

PDs	Sleep Disorder	Nervousness–Anxiety	Distress–Anger	Depression– Upset	Inferior to Others	Suicidal Thoughts
AEs						
Local pain	0.191	<0.001	0.004	<0.001	0.108	0.157
Fatigue	0.001	<0.001	<0.001	<0.001	0.002	0.003
Weakness	0.316	0.008	<0.001	<0.001	0.006	<0.001
Muscle pain	0.049	<0.001	0.001	<0.001	0.008	0.003
Headache	0.036	0.051	0.004	<0.001	0.014	0.003
Chills	0.092	0.055	0.006	<0.001	0.012	0.013
Dizziness	0.106	0.010	0.060	0.004	0.166	0.014
Local swelling	0.524	0.008	0.018	0.006	0.120	0.092
Joint pain	0.420	0.064	0.064	0.001	0.049	0.001
Poor appetite	0.097	<0.001	<0.001	<0.001	<0.001	<0.001
Fever	0.569	0.909	0.451	0.045	0.108	0.073

Footnote: Black cells indicate a significant association (*p* < 0.05); grey cells indicate a marginally significant association (*p*-values between 0.05 and 0.1); white cells indicate a non-significant association.

**Table 3 vaccines-11-00129-t003:** Multiple logistic regression of predicting factors (PFs) to preventive behaviors (PBs).

PBs	Avoiding Crowds	House Ventilated	House Sanitized	Hands Washed	Wearing Face Mask
PFs					
Age	1.01	0.99	0.99	0.97	0.96
Gender (Ref: male)	0.71	0.88	0.99	1.08	1.01
Occupation (Ref: Physician)					
Nurse	1.15	0.72	0.54	0.98	0.99
Others	1.24	0.88	0.67	1.41	1.64
Sleep disorders	0.68	1.34	0.96	1.19	1.21
Nervousness and anxiety	1.83	1.34	1.02	1.02	0.98
Distress and anger	0.93	1.14	0.75	0.56	0.72
Depression and upset	1.07	0.64	0.68	1.59	1.21
Inferiority to others	0.67	0.82	1.37	0.85	0.78
Suicidal thoughts	1.11	0.74	1.66	1.65	1.28

Footnote: Black cells indicate significantly more association with PBs; white cells indicate significantly less association with PBs; grey cells indicate a nonsignificant association.

## Data Availability

Corresponding authors will provide the data upon reasonable request.

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
