# Peer review of "Psychological Distress and Physical Adverse Events of COVID-19 Vaccination among Healthcare Workers in Taiwan"

_vaccines, 2023, doi:10.3390/vaccines11010129_

Round 1
Reviewer 1 Report
This study investigated the association between psychological distress and adverse effects of the Astra Zeneca Covid vaccine among Taiwanese healthcare workers (HCW); the study also investigated the association of psychological distress, and the behavior aimed at protection against Covid infection of HCW. The study was centered at one academic secondary/tertiary health care facility in Taiwan. Those healthcare workers, who presented for vaccination against Covid, were invited to participate, with a questionnaire, delivered online, which obtained from consenting participants, their demographics, psychological metrics aimed at determining psychological distress using a standardized questionnaire, capturing information at the time of receiving immunisation, their participation in protective behaviors aimed at reducing Covid infection, again at a single time point, and local and systemic adverse effects of the vaccine, serially up to 2 weeks after receiving the Covid vaccine.
The statistical method useful analysis of the data is indicated in the method section but is not explained in detail. About 10% of the total staff at the facility obtained the Covid vaccine. 75.6% of these subjects completed the questionnaire, and data from 483 individuals were analysed. The majority of the subjects, experienced local and systemic side effects of the vaccine, within the first three days at this declined to low levels and 14 days after immunisation. The range of adverse effects are similar to those reported among other immunised subjects, and there were no severe adverse effects.
The results of the multivariate statistical analysis showed that depression and distress-anger were significantly associated with physical adverse effects in the HCWs group , and there was no significant difference in these adverse effects between the different categories of healthcare workers.
The analysis of results also provided a detailed evaluation of the adherence to Covid infection preventative measures, among healthcare workers stratified by demographic measures as well as healthcare worker categories.
Although this study is based on a single healthcare center in Taiwan, the results of general interest, and method described could be used for similar studies elsewhere. The study provides insights into the relationship between prior mental health status and the occurrence of adverse effects to the Covid vaccine, and analyses differences between demographic groups studied, as well as comparison of results across different categories of healthcare workers.
The generalizability of these results is limited by the study being limited to single center, possible ascertainment bias, induced by the likelihood of those who accept immunisation against Covid, contrasted with the large majority who don't volunteer to be immunised, and the fact that the assessment of mental health parameters, is carried out only at one time point, when the subjects presented themselves for immunisation.
Comments to authors:
1. The brief description of the method of statistical analysis, [“Descriptive statistics were used to analyze the participants' characteristics including frequency and means. We used generalized estimating equations (GEE) with a linear scale function and exchangeable structure correlation matrix to analyze the relationship between psychological disorder and physical AEs after receiving the COVID-19 vaccine”], may not be fully understood by the general reader. It would be helpful to include a paragraph paraphrasing the method in more detail.
2. The data presented in table 2 is difficult for the reader to assimilate. I would suggest that this comprehensive date is added as a supplementary table and the most significant information from here is presented in a pictorial fashion, if possible.
3. The data presented in table 3 is difficult for the reader to assimilate. In my opinion it would be easier to comprehend if the most significant data could be presented in a pictorial fashion, with the comprehensive table being included as a supplementary table
4. For the data described in this paragraph: ‘There was a significant difference between male and female HCWs who experienced AEs. Similarly, in the comparison by age group, there was a significant difference in the incidence of AEs, except for the incidence of local swelling (p = 0.094). There was no significant difference between the type of health professionals and the incidence of AEs after receiving the COVID-19 vaccine’, it would be helpful if the absolute values were included.
5. The generalizability of these results is limited by possible ascertainment bias, induced by the likelihood of those who accept immunisation against Covid, contrasted with the large majority who don't volunteer to be immunised, with respect to the evaluation of the adherence to Covid infection preventative measures, among healthcare workers.
Reviewer 2 Report
The paper reports an interesting connection between the side effects of vaccination, psychological adverse effects, and the adoption of protective behaviors. Although the sample size is limited to about 10% of the healthcare workers in one hospital in Taiwan (<500 individuals), the results are interesting. The paper is mostly well-written, and the statistics are solid. This is probably a good fit for vaccines after making a few small changes:
1) L45: "The prevalence of COVID-19 among frontline healthcare workers (HCWs) was 2747 cases per 100,000 compared with 242 cases per 100,000 people in the general community [1] " Please clarify that this is during the first months of the pandemic, March-April 2020. This is definitely not true right now.
2) L56: There are only 483 individuals who fully completed the survey during March-April 2021. This is a very small sample out of the 5948 healthcare workers in the hospital. Did the survey extend past these months? Are the 483 participants those who completed all time points of the survey? What robustness checks were used to ensure the 483 is a representation of the 5983 individuals? This needs to be clarified in the paper. The content of Line 312 needs to be expanded to explain how these limitations on the analysis can "tolerate" missing data (the word tolerate is strangely used).
3) L242: "Therefore, to reduce the potential AEs after vaccination in younger people and women, it can be achieved by giving 243 a lower COVID-19 vaccine dose [26]." The paper does not conclude this. This is very prescriptive, and your paper, at best, works with correlations. Remove this comment, as this is not the conclusion of your paper.
4) This paper has a lot of overlap with the paper [29], and you really need to explain better in the context of Taiwan healthcare workers. What are the main differences with [29]? In general, better connections with the literature are needed. The paragraph in Line 245 is important and needs to be expanded and broken down.
5) L267: Rakel needs to be edited
6) L353 w should be W
Reviewer 3 Report
It can be improved by adding some text about the sample frame.
